Integrated lipidomic and transcriptomic analysis reveals diacylglycerol accumulation in olive of Longnan (China)

Hu Wei 1
Ma Junyi 1 skymjy@nwnu.edu.cn
Zhang Hongjie 1
Miu Xin 1
Miao Xin 1
Deng Yu 2
1 College of Life Science, Northwest Normal University , Lanzhou, Gansu , China
2 Institute of Olive, Longnan Academy of Economic Forestry , Wudu, Gansu , China
Irfan Mohammad
Electronic publication date: 2023 Aug 11
Publication date: 2023
Volume: 11
Electronic Location ID: e15724
Received 2023 Mar 15; Accepted 2023 Jun 18
Copyright: © 2023 Hu et al.
Copyright year: 2023
Copyright holder: Hu et al.
License: This is an open access article distributed under the terms of the Creative Commons Attribution License, which permits unrestricted use, distribution, reproduction and adaptation in any medium and for any purpose provided that it is properly attributed. For attribution, the original author(s), title, publication source (PeerJ) and either DOI or URL of the article must be cited.
License URL: https://creativecommons.org/licenses/by/4.0/

Keywords: Diacylglycerols accumulation, Gene expression, Lipidomics, Transcriptomics, Virgin olive oil

Funding: Basic Research Innovation Group Program Project of Gansu Province 1506RJIA116 Major Research Projects Incubation Programme of Northwest Normal University in 2022 NWNU-LKZD2022-02 This research was financially supported by the Basic Research Innovation Group Program Project of Gansu Province (1506RJIA116) and the Major Research Projects Incubation Programme of Northwest Normal University in 2022 (NWNU-LKZD2022-02). The funders had no role in study design, data collection and analysis, decision to publish, or preparation of the manuscript.

==============================
Background

Olive (Olea europaea L.) oil accumulate more diacylglycerols (DAG) than mostly vegetable oils. Unsaturated fatty acids-enriched DAG consumption enhanced wellness in subjects. However, the mechanism of DAG accumulation is not yet fully understood.

Methods

In this study, gene network of DAG accumulation and fatty acid composition in the two olive mesocarps (“Chenggu 32” (CG) and “Koroneiki” (QJ)) were investigated by integrating lipidome and transcriptome techniques.

Results

A total of 1,408 lipid molecules were identified by lipidomic analysis in olive mesocarp, of which DAG (DAG36:3, DAG36:4 and DAG36:5) showed higher content, and triacylglycerols (TAG54:3, TAG54:4) exhibited opposite trend in CG. Specifically, DAG was rich in polyunsaturated fatty acids (especially C18:2) at the sn-2 position, which was inconsistent with TAG at the same positions (Primarily C18:1). Transcriptomic analysis revealed that phospholipase C (NPC, EC 3.1.4.3) were up-regulated relative to QJ, whereas diacylglycerol kinase (ATP) (DGK, EC 2.7.1.107), diacylglycerol acyltransferase (DGAT, EC 2.3.1.20), and phospholipid: diacylglycerol acyltransferase (PDAT, EC 2.3.1.158) were down-regulated.

Conclusion

We speculated that the non-acyl coenzyme A pathway played a significant role in DAG biosynthesis. Additionally, fatty acyl-ACP thioesterase B (FATB, EC 3.1.2.14), stearoyl [acyl-carrier-protein] 9-desaturase (SAD, EC 1.14.19.2) and omega-6 fatty acid desaturase (FAD2, EC 1.14.19.6) were highly expressed in CG and may be involved in regulating fatty acid composition. Meanwhile, phospholipase A1 (LCAT, EC 3.1.1.32) involved in the acyl editing reaction facilitated PUFA linkage at the sn-2 position of DAG. Our findings provide novel insights to increase the DAG content, improve the fatty acid composition of olive oil, and identify candidate genes for the production of DAG-rich oils.

Introduction

Naturally occurring vegetable oils and fats are vital ingredients in the human diet, mainly composed of triacylglycerols (TAG), followed by diacylglycerols (DAG) and monoacylglycerols (MAG). Research shows that regular consumption of TAG-rich oils increases the risk of obesity and numerous chronic diseases (Prabhavathi Devi et al., 2018). DAGs consist of a glycerol backbone esterified with two fatty acids, divided into two isoforms: 1,2-DAG and 1,3-DAG (in a 7:3 ratio) (Lee et al., 2019). Consumption of DAG-rich oils, which rarely accumulate in the body, is mainly attributed to different metabolic pathways compared with TAG-rich oils, making it difficult to be absorbed. Evidence shows that increasing ingestion of DAG not only regulates blood lipid levels and inhibits weight gain but also alleviates diabetes and its complications while playing a role in regulating metabolic syndrome (Prabhavathi Devi et al., 2018; Saito et al., 2010; Yanai et al., 2007). Qualitative and quantitative studies suggest that the intake of DAG oil is safe for humans, and its bioavailability is similar to that of TAG oil (Taguchi et al., 2001; Morita & Soni, 2009). Due to numerous health benefits, DAG-rich oils have found extensive applications in the food industry (Lee et al., 2019).

The primary process of DAG biosynthesis involves de novo biosynthesis of fatty acids (FAs) in the plastid and their assembly in the endoplasmic reticulum (ER) (Ohlroggeav & Browse, 1995). This process involves a string of enzymes encoded by functional genes. Acetyl-coenzyme A carboxylase carboxyl transferase subunit alpha (accA) catalyzes the formation of malonyl-CoA from acetyl-CoA, which is the first step in fatty acid biosynthesis. The FAs synthase complex is responsible for extending the FA chain to 16:0-ACP and/or 18:0-ACP. Glycerol-3-phosphate acyltransferase (GPAT) serves as the starting point for DAG biosynthesis, affecting the binding of glycerol and FAs. Diacylglycerol O-acyltransferase (DGAT) (Zheng et al., 2008) and phospholipid: diacylglycerol acyltransferase (PDAT) have overlapping effects in promoting the biosynthesis of TAG (Fan et al., 2014; Kim et al., 2011). Furthermore, some genes have been shown to regulate the composition of FAs. Stearoyl [acyl-carrier-protein] 9-desaturase (SAD) catalyzes the desaturation of C18:0-ACP to C18:1-ACP within the plastid, while delta(12)-fatty-acid desaturase FAD2-like (FAD2) further catalyzes the desaturation of C18:1-PC to C18:2-PC in the ER. Both enzymes affect the FA composition of oil crops (Chen et al., 2015; Bates, Stymne & Ohlrogge, 2013).

Olive (Olea europaea L.), one of the notable oil fruit crops, originates from the Mediterranean basin. It was introduced into Longnan city in 1975, where it serves as an olive oil demonstration base in China. The mesocarp of olive fruits is mainly responsible for producing oil, accounting for approximately 28% of the fresh fruits weight (Guodong et al., 2019). Virgin olive oil (VOO) extracted through physical pressing process are rich in polyphenols, flavones, squalene and vitamin E, etc. These chemical compositions are beneficial to health, such as anti-cancer, anti-inflammation and anti-diabetes (Raederstorff, 2009; Shah et al., 2017). Concurrently, polyunsaturated fatty acids (PUFAs), primarily oleic acid, which are present in high levels in VOO, can enhance digestive system function, reduce the incidence of cardiovascular disease and promote the development of skeletal and nervous systems (Visioli & Galli, 2002; Caramia et al., 2012; Picklo et al., 2017). Therefore, VOO is known as the “queen of vegetable oils” due to its high reputation (Alagna et al., 2016; Weng & Yen, 2012). The quality of VOO is the result of several factors including cultivar, climate and orchard management, etc. (Mannina et al., 2001). Notably, the proportions of the DAG in the VOO is higher than that in the majority of edible vegetable oils, with values of approximately 5.5% (Morita & Soni, 2009). Currently, many researchers are attempting to enzymatically or chemicall synthesize DAG-rich oils through various methods. However, impurities still exist in the manufacturing process, so it is important to improve the purity of DAG (Lee et al., 2019). Olive oil is nutrient-dense and changing the composition of VOO and increasing DAG content through molecular breeding and biotechnology strategies are necessary for further improving the quality of oil.

Lipidome, a bridge between genotype and phenotype, elucidates lipid metabolisms at the cellular level by studying the composition, structure and quantification of lipids in biological samples. Transcriptomic can effectively identified and quantify gene expression levels in plants. In recent years, lipidomic and transcriptomic sequencing have been used to investigate the molecular mechanisms of lipid composition in soybean, rapeseed, tung tree and peanut (Woodfield et al., 2018; Zhang et al., 2018, 2020a, 2020b), demonstrating considerable progress in lipid turnover, oil accumulation, growth and development and physiological regulation. We aimed to elucidate the potential molecular mechanism of DAG accumulation which rich in PUFAs. Two varieties of olives from the same origin were selected for lipidome and transcriptome sequencing. Crucial genes, such as diacylglycerol kinase (ATP) (DGK), phosphatidate phosphatase 1 like (DPP1), non-specific phospholipase C (NPC), phospholipase A(1) LCAT3-like (LCAT3), fatty acyl-ACP thioesterase B (FATB), OePDAT, OeSAD and OeFAD2, were analyzed. Our study provided a basis for breeding olive varieties that are abundant in PUFAs and DAGs.

Materials and Methods

Plant material

Two varieties of olives, “Chenggu 32” (CG) and “Koroneiki” (QJ) were used as raw materials for transcriptome and lipidome detection. The olives were collected from the Olive Germplasm Research Garden in Longnan city, Gansu province of China (altitude: 1,036–1,048 m; longitude: 104°53′30″E, latitude: 33°24′03″N; average temperature: 15.3 °C; maximum temperature: 38.2 °C; minimum temperature: −7.0 °C; relative humidity: 56.6%; annual precipitation: 468.0 mm; annual sunshine hours: 1,871 h; soil type: sandy loam with a pH of 7.9). Samples were collected at the fourth stage of olive fruit maturity in 2020, specifically on October 20th and 30th. The same evaluator subjectively evaluated the color change of the olive peel, and harvesting occurred when more than half of the olive peel had turned reddish-brown (Yousfi, Cert & Garcia, 2006). Each sample had three biological replicates, resulting in a total of six groups. After collection, the samples were immediately frozen in liquid nitrogen and stored at −80 °C for further studies.

Oil content and fatty acids composition analysis

Olive oil was extracted using an Abencor® System (MC2 Ingenieríay Sistemas S.L, Sevilla, Spain). The extraction process were as follows: 800.0 g of fresh olive fruit were crushed and degreased at 30 °C for 1 h. They were then centrifuged twice at 25 °C for 60 s each time with a rotational speed of 5,000 r/min. After centrifugation, the oil phase and water phase were transferred to a measuring cylinder of 250.0 mL and diluted to the mark with deionized water at 25 °C. After resting, the volume of oil phase was measured and recorded. Then, the oil phase was transferred into a glass bottle, sealed and stored at 4 °C. Each sample was repeated three times biologically. The oil content = (V × 0.915)/m × 100% (V: the volume of oil (cm3), m: initial pulp mass (g)). The methods for preparing fatty acid (FA) methyl esters were the same as those in a previous report (Lü et al., 2021). The fatty acids composition and relative percentage of olive oils were qualitatively and quantitatively analysed by GC-MS. Quantitative analysis was based on NIST 2011 mass spectrometry database and peak area normalization method. The GC-MS analysis were performed three times.

UPLC-Q exactive/MS-based lipidomic analysis

Lipids were extracted using the following method: Approximately 30 mg of mesocarp was ground in liquid nitrogen, followed by the addition of an internal lipid standard mixture (20 μL, Table S2), MTBE (800 μL) and pre-cooled methanol (240 μL) for each sample. The mixture was vortexed for 20 min at 4 °C and then left to stand for 30 min at room temperature. The solution was centrifuged at 14,000×g for 15 min at 10 °C. The upper organic phase was extracted and dried with gaseous nitrogen. The lipid extracts were redissolved in a mixture of 90% isopropanol/acetonitrile (200 μL) and then centrifuged again for an additional 15 min at 10 °C. Finally, the supernatant fluid was retained for analysis.

Samples were separated using UPLC Nexera LC-30A ultra performance liquid chromatography system (Shimadzu, Kyoto, Japan) with a C18 column (2.1 mm × 100 mm, 1.7 µm, Milford, MA, USA) at 45 °C. The mobile phases included A: an acetonitrile aqueous solution (acetonitrile/water = 6/4, V/V) and B: an acetonitrile isopropanol solution (acetonitrile/isopropanol = 1/9, V/V). The gradient elution process proceeded as follows: from 0 to 2 min, 30% B; from 3 to 25 min, B increased linearly from 30% to 100%; from 26 to 35 min, the concentration of B decreased to 5%. After that, the separated substances were analyzed using a Q-Exactive mass spectrometer (Thermo Scientific, Waltham, MA, USA) and detected from 200 to 1,800 m/z based on the following conditions: source temperature of 300 °C; capillary temperature of 350 °C; spray voltage of 3.0 KV; S-Lens RF level at 50%; sheath gas flow rate at 45 arb, aux gas flow rate at 15 arb and sweep gas flow rate at 1 arb. Each group conducted six biological replicates.

Lipidsearch software version 4.2 (Thermo Scientific, Waltham, MA, USA) was used to extract and identify the peaks of lipid molecules and internal standard lipid molecules. The acquired data were imported into SIMPCA-P 16.1 (Umetrics, Umea, Sweden) for multivariate statistical analysis, which included principal component analysis (PCA) and orthogonal partial least squares discriminant analysis (OPLS-DA). Lipids with significant differences were identified by VIP > 1 and P-value < 0.05. Cluster 3.0 (http://bonsai.hgc.jp/~mdehoon/software/cluster/software.htm) and Java Treeview software (http://jtreeview.sourceforge.net) were used for the lipid expression data for hierarchical clustering analysis, as well as to determine the correlation between variables based on the Pearson correlation analysis.

Transcriptomic sequencing and analysis

Total RNA was extracted from olive mesocarps using Qiagen RNeasy® Mini Plant Kit (Valencia, CA, USA). The libraries were prepared according to the ABclonal mRNA-seq Lib Prep Kit (ABclonal, Wuhan, China) and the quality of the libraries was evaluated using the Agilent Bioanalyzer 4150. Finally, sequencing was performed on the Illumina Novaseq 6000/MGISEQ-T7 sequencing platform from Shanghai Applied Protein Technology Co, Ltd. Each group performed three biological repeats. The raw reads obtained by RNA-seq were filtered out of low-quality reads, adapter reads and sequences with N ratios more than 10% to obtain clean reads that could be used for subsequent analysis. The software HISAT2 (http://daehwankimlab.github.io/hisat2/) was used to compare the clean reads to the olive reference genome (http://plants.ensembl.org/Olea_europaea/Info/Index). FPKM, which represents the expression level of transcript, was calculated based on the software FeatureCounts (http://subread.sourceforge.net/) and (http://subread.sourceforge.net/). Differential expression analysis was performed using DESeq2 (http://bioconductor.org/packages/release/bioc/html/DESeq2.html) and the default screening thresholds were |log2FC| > 1 and Padj < 0.05. Gene ontology (GO) function enrichment and kyoto encyclopedia of genes and genomes (KEGG) pathway enrichment analyses were performed by the clusterProfiler R package. Differentially expressed protein with FDR < 0.05 were considered. GO function annotation was based on Blast2GO (https://www.blast2go.com), KEGG function annotation was based on Kobas (https://bio.tools/kobas).

Quantitative reverse transcription PCR analysis

Total RNA was extracted utilizing the KKFast Plant RNApure kit (Zoman Biotech, Beijing, China). Total RNA was reverse transcribed to cDNA using the PrimeScript RT Reagent Kit (TransGen Biotech, Beijing, China). Quantitative reverse transcription PCR (qRT-PCR) was performed by Taq Pro Universal SYBR qPCR Master mix (Vazyme, Nanjing, China) on the CFX96 Touch System (Bio-Rad, Hercules, CA, USA). The EF1-alpha (XM_002527974.1) was used as the internal reference gene. The primers used in the study were designed using the program Primer3Plus (https://www.bioinformatics.nl/cgi-bin/primer3plus/primer3plus.cgi) and were listed in Table S2. Finally, we used the 2−ΔΔCt method to calculate the relative expression levels of the genes.

Statistical analyses

A student’s t-test was performed by IBM SPSS 26 (SPSS Inc, Chicago, IL, USA). P < 0.05 was used as the significance level. Our results of the analysis are shown in Origin 2021 (OriginLab, Northampton, MA, USA).

Results

Oil accumulation in olive mesocarp

Two varieties, “CG” and “QJ” with same growing condition were selected as experimental samples for total lipids and FAs analysis. We first compared the data distribution of each variate between CG and QJ groups, using the student’s t-test (normal distribution). P < 0.05 was used as the significance level. We observed significant difference in the VOO contents of mesocarp, which decreased by 55.67% in CG group relative to QJ group (Fig. 1A). The nutritional value of VOO was influenced by the ratio of SFA/UFA (Carlsson et al., 2011). FAs composition were examined by GC-MS technique. Nine FAs were tested by GC in the VOO, ranging from C16 to C20 carbon atoms. The principal FAs found in VOO were palmitic acid (C16:0), stearic acid (C18:0), oleic acid (C18:1) and linoleic acid (C18:2). Compared to QJ group, the saturated fatty acids (SFAs) content and polyunsaturated fatty acids (PUFAs) content increased in the CG (by 21.88% and 63.83%, respectively), whereas the Monounsaturated fatty acid (MUFAs) content decreased (by 27.08%). Particularly, C18:1 content decreased by 28.07% and C18:2 content decreased by 63.44% in QJ (Figs. 1B and 1C, Tables S3 and S4).

Figure 1 The virgin olive oil content and relative content of fatty acid difference between CG and QJ.

Oil content of olive fresh fruit at picking time (A). Relative content of saturated fatty acid, monounsaturated fatty acid, and polyunsaturated fatty acid in olive fresh fruit. Error bars indicate standard deviation. The asterisk symbol (*) is used to indicate significant differences between CG and QJ, according to student’s t-test (*P < 0.05, **P < 0.01) (B). Prime fatty acids composition of olive oil (C).

Lipid component of olive mesocarp exhibits differences

The olive mesocarps was performed by lipidome in UPLC-Q Exactive/MS technique to clarify the quantitative difference between CG and QJ. We thoroughly analyzed the stability of the instrument, the reproducibility of experiments and the data reliability of quality, showing our data were valid for subsequent analysis. A total of 38 lipid classes (three glycerolipids, 14 glycerophospholipids, nine sphingolipids, three serol lipids, one prenol lipid, six saccharolipids, two fatty acyls) and 1,413 lipid species were identified (Table S5). Principal component analysis (PCA) of sample was separated into two groups, corresponding to CG and QJ (Fig. 2A), and hierarchical cluster analysis indicated the same results (Fig. S3). Olive mesocarp contained significant levels of ChE and DGDG but only traced amounts of GM1, SM and LPI (Figs. S2A and S2B). Additionally, we found that DAG content (13,975.42 µg/g) increased by 58.61% and TAG content (13,500.94 µg/g) decreased by 34.35% in the CG group relative to the QJ Group (Fig. 2B). Collectively, we revealed lipid differences between two olive varieties.

Figure 2 PCA map, volcano plot, differentially expressed genes (DEGs) and gene ontology (GO) enrichment in CG and QJ.

PCA map of three biological replications for two varieties (A). The volcano plot shows the distribution of differentially expressed genes (DEGs) in the two samples. The red points represent the up-regulated DEGs, the green points represent the down-regulated DEGs, while the black ones represent the unchanged genes (B). Gene ontology (GO) enrichment analysis of DEGs. The abscissa represents the GO classification, the ordinate represents the number of DEGs annotated to the terms (C). KEGG enrichment analysis based on the most abundant pathways of DEGs (CG vs QJ). The size of each point represents the number of genes (D).

Lipid molecules differences for TAG and DAG in olive mesocarp

We put to use OPLS-DA to recognized differential lipid molecules in two varieties olive mesocarps, on the ground of VIP ≥ 1 and P value ≤ 0.05. The consequences indicated 14 up-regulated and 19 down-regulated lipids (Fig. S3). Especially, there was a significant accumulated of DAGs in CG, particularly DAG 36:3, DAG 36:4 and DAG 36:5. Interestingly, general up-regulated DAGs bore PUFAs such as C18:2 and C18:3 at the sn-2 position, while MUFAs, such as C18:1, mainly concentrated in TAG. However, TAGs were highly accumulated in QJ, mainly TAG54:3 and TAG54:4, and the sn-2 and/or sn-3 positions were predominantly C18:1 (Figs. 2C–2E, Table S6). DAG acts as the precursor for TAG biosynthesis, whereas the FAs component of DAGs differ with TAGs but similar to PCs (PC 36:3 and PC 36:4). DAG production, catalyzed directly by PA, but FAs of the two lipid molecules exhibit significant differences. Our research discovered that the UFAs prepared for DAG biosynthesis in olive generated from diverse sources including Kennedy pathway, in addition to phospholipid-dependent non-acyl CoA pathway.

Gene expression and functional classification in olive mesocarp

To investigate the pathway of DAG biosynthesis and difference for the FAs content, we performed transcriptome sequencing on olive mesocarps using the RNA-seq technique. A total of 35.78 GB clean reads were obtained, with Q2 and Q3 being 97.37% and 92.32%, indicating that our results were reliable for further analysis (Table S7). We detected 10,114 genes in olive mesocarp with |log2FC| > 1, FDR < 0.01. Differential expression analysis revealed that 4,619 up-regulated and 5,495 down-regulated genes in CG, compared to QJ (Fig. 3B, Table S8). PCA score plots showed that the contribution of the PC1 and PC2 were 63.93% and 9.19%, respectively (Fig. 3A), demonstrating a sharp distinction between CG and QJ group. Pearson correlation analysis and hierarchical cluster analysis (HCA) of the samples between the two groups confirmed the PCA results (Figs. S4A and S4B).

Figure 3 The major profiles of expression of metabolites in the CG and QJ.

Principal component analysis (PCA) enrichment analysis of transcripts (A). Prime lipidlon in fresh olive mesocarp (B). Heatmap demonstrates differential level of lipid molecules and for each one of them in the ratio between CG and QJ transformed into log2 and depicted with color scale (from yellow to green) (C). The content of significant difference lipid compounds in PA and PC (D). The content of significant difference lipid compounds in DG and TG (E). Abbreviations: PA, phosphatidic acid; PC, phosphatidyl-choline; DAG, diacylglycerol; TAG, triacylglycerol. Error bars indicate standard deviation. The asterisk symbol (*) indicate significant differences between CG and QJ (*P < 0.05, **P < 0.01, ***P < 0.001).

GO classified differentially expressed genes (DEGs) into three categories, including 30 subcategories. Among them, biological processes (BP) were the most abundant (18), molecular functions (MF) were the second (10), and cellular components (CC) were the least (two). The results showed that the BP-related DEGs were mainly concentrated in metabolic processes (102 DEGs, 54 down-regulated) and organic matter metabolic processes (87 DEGs, 44 down-regulated). The CC-related DEGs were mostly absorbed in binding (97 DEGs, 48 down-regulated) and catalytic activity (93 DEGs, 51 down-regulated). The molecular function-related DEGs were primarily focus on the cellular fraction (126 DEGs, 67 down-regulated) and cellular (112 DEGs, 67 down-regulated) (Fig. 3C). An enrichment analysis of KEGG pathways was performed to further investigate the role of DEGs in these pathways (Fig. 3D). The results showed that DEGs were significantly enriched in metabolic pathways, biosynthesis of secondary metabolites pathways and photosynthesis pathways. We noted that DEGs were also significantly enriched in pathways associated with lipid accumulation: fatty acid biosynthesis, glycerolipid metabolism, glycerophospholipid metabolism, linoleic acid metabolism, biosynthesis of unsaturated fatty acids (Table S9).

Identification of differentially expressed genes involved in DAG biosynthesis

We studied on the lipid metabolism of olive mesocarps, as well as identified 161 DEGs by transcriptomic analysis involved in oil accumulation (Table S8).

The glycolysis pathway is responsible for breaking glucose into pyruvate for fatty acid biosynthesis. We found that the hexokinase-1-like (HK) genes were down regulated in CG by 2.44-fold. ATP-dependent 6-phosphofructokinase 6-like (pfkA), encoding critical enzyme, significantly decreased by 9.13-fold in CG compared to QJ. Simultaneously, the fructose-bisphosphate aldolase (ALDO) and glyceraldehyde-3-phosphate dehydrogenase (GAPDH) genes were reduced in CG than QJ by at least 2–6-fold. Other downstream genes involved in the glycolysis pathway, including 2,3-bisphosphoglycerate-independent phosphoglycerate mutase (gpmI) were down-regulated in CG by around 2-fold, resulting in pyruvate decrease (Fig. 4A).

Figure 4 Changes in the major metabolites and genes involved in the diacylglycerol biosynthesis pathway.

Interaction relationship in the major metabolites and genes involved in glycolysis (A, left), citrate cycle (A, right), fatty acids biosynthesis (B, left), glycerol and glycerophospholipid metabolism (B, right) metabolism pathway. Significant differences in genes expression levels are indicated by the intensity of colors. Red or blue gene symbols indicate up-regulated or down-regulated DEGs, respectively. Abbreviation of genes used in figures are supplemented in Table S1. Thermal diagram of the correlation coefficient of DEGs and lipids in the olive mesocarp (C). Abbreviations for genes are as follows: hexokinase(HK), 6-phosphofructokinase (pfkA), fructose-bisphosphate aldolase (ALDO), glyceraldehyde-3-phosphate dehydrogenase (GAPDH), 2,3-bisphosphoglycerate-independent phosphoglycerate mutase (gpml), isocitrate dehydrogenase (NADP (+)) (IDH1), acetyl-CoA carboxylase(accA), fatty acyl-ACP thioesterase B (FATB), stearoyl-[acyl-carrier-protein] 9-desaturase (SAD), glycerol-3-phosphate acyltransferase (GPAT), lysophosphatidate acyltransferase (AGPAT), phosphatidate phosphatase (DPP1), diacylglycerol kinase (ATP) (DGK), diacylglycerol O-acyltransferase (DGAT), omega-6 fatty acid desaturase (FAD2), phospholipase C (NPC), phospholipase A1 (LCAT), phospholipid: diacylglycerol acyltransferase (PDAT).

Pyruvate formed from the glycolysis pathway, is catalyzed by the pyruvate dehydrogenase E2 component (DLAT) to form the acetyl-CoA, which is the committed step in citrate cycle pathway. In particular, OeDLAT gene was expressed higher in CG by 2.12-fold, compared to QJ. Strikingly, expression levels of isocitrate dehydrogenase (NADP) (IDH1), 2-oxoglutarate dehydrogenase (OGDH), and 2-oxoglutarate dehydrogenase E2 component (dihydrolipoamide succinyltransferase) (DLST) were all increased by 2–3-fold in CG compared to QJ, promoting succinyl CoA biosynthesis (Fig. 4A).

FAs are essential components to form glycerides. Within the plastid, FAs biosynthesis begins with the unreversible catalysis to acetyl-CoA by accA to yield malonyl-CoA. Then, acetyl-CoA plays a role in building cornerstone in a continuous polymerization reaction to form FAs. We found that the transcript levels of OeaccA and OeFATB genes were enhanced in QJ by 2–4-fold, suggesting partly associated with FAs biosynthesis and mechanism of acyl CoA export. Notably, OeSAD genes were expressed higher in QJ by 2.60-fold higher compared to CG, corresponding with the obviously increased levels of C18:0 in QJ (Fig. 4B).

Within the ER, FAs are esterified in turn at the sn-1 and sn-2 positions of glycerol-3-phosphate by GPAT and acyl phosphate: glycerol-3-phosphate acyltransferase (LPAT) to generate lysophosphatidic acid (LPA) and phosphatidic acid (PA). Afterwards, the phosphate at the sn-2 position is dephosphorylated by phosphatidic acid phosphatases (PAPs) or other phospholipids to yield sn-1,2-diacylglycerol. Then, TAG is acylated at the sn-3 position by acyl-CoA molecule using DGAT to produce TAG. Notably, DAG could also be acylated using phosphatidylcholine (PC) as the acyl donor by PDAT in an acyl-CoA-independent process to generate TAG. The non-specific phospholipase C (NPC) is involved in PC-DAG conversion and plays a role in acyl transferring reactions. Meanwhile, DAG could reversely produce PA catalyzed by DGK, which decreases the neutral lipids DAG content. We observed that genes related to DAG biosynthesis, such as OeGPAT, 1-acyl-sn-glycerol-3-phosphate acyltransferase (AGPAT), OeDPP1 and OeDGAT were well in line with the decreased levels of TAG in CG. However, the DAG content was higher in CG compared to QJ. In contrast to above genes expression trend, expression levels of OeNPC5 were up-regulated by 18.16-fold, while OeDGK7 were down-regulated by 2.04-fold, consistent with the higher level of DAG in CG.

To understand the reasons for higher PUFA content of DAG in CG, we further investigated the acyl editing reactions affecting DAG accumulation. PC transforms C18:1 linked by esterification into C18:2 catalyzed by FAD2, which is essential for C18:2 biosynthesis in oil seeds (Bates, Stymne & Ohlrogge, 2013). The OeFAD2 genes were expressed as much as 6.72-fold higher in CG compared with QJ. In order to explore the relationship between differentially expressed genes and differential lipid molecules affecting DAG accumulation. We carried out Pearson correlation analysis. The Pearson correlation coefficient is between −1 and +1. Among them, r > 0 means positive correlation, r < 0 means negative correlation. The results indicated that the expression of OeNPC5 and OeLCAT3 (increased by 4.18-fold) were positively correlated with DAG content, whereas negatively correlated with OeaccA, OeGPAT, OeAGPAT and OeDPP1 contents (Figs. 4B and 4C).

Gene validation by qRT-PCR analysis

To verify the DEGs obtained from transcriptome sequencing, eight DEGs related to DAG biosynthesis and FAs composition pathways were selected to experience qRT-PCR verification. We found that four genes were decreased in CG and four genes was expressed higher in CG. Genes expression trend were consistent with the RNA-seq analysis, indicating that the sequencing results from RNA-seq were reliable for further research (Fig. 5).

Figure 5 qRT-PCR results of the expression of genes related to diacylglycerol synthesis.

Quantitative reverse transcription PCR analysis of eight significant difference genes. Error bars indicate standard deviation. The lowercase letters indicated significant differences (P < 0.05). Abbreviations for genes are shown in Table S2.

Discussion

In our study, lipidome and transcriptome experiments were performed to explore DAG accumulation mechanism in “CG” and “QJ”, with varying VOO content. Then the results were compared. A total of 1,408 differentially expressed lipid molecules were detected in the lipidomic analysis, while 35.78 Gb of clean reads and 10,115 DEGs were obtained from transcriptome sequencing. Correlation relationships between lipid content, FAs composition and genes expression were explored by omics joint analysis. Our results provided a foundation for improving the quality of VOO and researching DAG-enriched functional olive oil.

The majority of neutral lipids in plants were stored as TAG. We found that the VOO and TAG content in CG were lower compared to QJ. However, the DAG content of oils from the two cultivars showed opposite trends, especially DAG36:3 and DAG36:4. The GC-MS analysis revealed that the fatty acid components of two olive cultivars had identical compositions. This suggests that QJ had notably accumulated MUFA (C18:1), while the content of PUFA (especially C18:2) showed a negative correlation with the accumulation of MUFA. Meanwhile, differential lipid molecular analysis revealed that the majority of the PUFA (C18:2) in CG were located at the sn-2 position in DAG. PUFAs provide a vast array of benefits, ranging from health enhancements to protection against inflammation and illness, such as maintaining bone health, regulating metabolism, reducing cholesterol and fighting diabetes (Lee et al., 2019; Li-Beisson et al., 2013). Since C18:2 and C18:3 cannot be natural synthesis by human beings, both are necessarily acquired through plant and/or animal oils in the daily diet.

The glycolysis pathway in the olive mesocarp is mainly restricted by two rate-limiting reactions. The reaction catalyzed by HK is the primary pathway for hexoses to be incorporated into metabolism (Guglielminetti et al., 2000). OeHK, the first rate-limiting enzyme in the glycolytic pathway, was found to be down-regulated in CG. These findings may have implications for assessing glycolytic flux and its regulation in olive oil metabolism (Lai et al., 1999).

accA is the first rate-limiting enzyme in the biosynthetic pathway of FAs. Research has shown that overexpression of accA not only increased the oil content of brassica napus but also affects the composition of FAs by promoting C18:1 biosynthesis (Weselake et al., 2009). The genes encoding OeaccA exhibited lower expression levels, which consist with the VOO content in CG compared to QJ. This indicates that the OeaccA genes may influence FA accumulation in olive mesocarps. GPAT is involved in the first step of the glycerolipid biosynthesis pathway, which has been considered to play a rate-limiting role in the generation of lysophosphatidic acid (LPA). Additionally, GPAT selectively esterifies 18:1-ACP or 16:0-ACP at the sn-1 position of glycerol-3-phosphate (Murata & Tasaka, 1997) and its expression level positively affects oil content and promotes glycerolipid biosynthesis (Singer et al., 2016). The expression level of OeGPAT genes were consisted with higher oil content in QJ (Table S3). The DPP1 genes regulates lipid metabolism by managing the cellular levels of its substrate PA and product DAG, which are necessary for organism growth, development and stress response to adversity (Munnik, 2001; Testerink & Munnik, 2005). The results indicate that the expression values of genes above mentioned were decreased, in contrast to the increasing DAG content in CG. We speculate that the Kennedy pathway may not be the primary route leading to DAG accumulation in olives.

PC plays a role in decorating and transferring FAs to TAG in oilseeds, serving as a transfer station (Weselake et al., 2009). Additionally, most of the FAs that emerge from the plastid as acyl-CoA enter into PC instead of following the traditional Kennedy pathway (Allen, Bates & Tjellstrom, 2015; Bates et al., 2009). In the PC-centred acyl trafficking reaction, acyl groups are transferred from the acyl-CoA pool to the PC via catalysis by LCAT (Weselake et al., 2009; Chapman & Ohlrogge, 2011). Then, the acyl groups on PC are modified by desaturation, epoxidation, conjugation or hydroxylation processes catalyzed by FAD2-like enzymes (Carlsson et al., 2011). Subsequently, the modified acyl group undergoes three transfer reactions: Acyl groups were input into the DAG pool through the catalytic action of phospholipase C (PLC) (Slack et al., 1985; Weselake et al., 2009; Lu et al., 2009); Most acyl groups at the sn-2 position are mediated by PDAT to form TAG and lysophospholipid (LPC) from PC (Dahlqvist et al., 2000); Acyl groups enter the acyl-CoA pool through forward/reverse processes mediated by LCAT3 for lipid biosynthesis (Jessen et al., 2015; Lager et al., 2013). In our study, all of the above genes were up-regulated in CG, which was consistent with the high content of DAG.

In plants, PLCs are classified into two categories based on their substrate specificity: specific PLCs hydrolyze phosphatidylinositol, while non-specific PLCs cleave common phospholipids (Weselake et al., 2009). In our study, the expression level of OeNPC was found to be higher in CG than in QJ. Previous research revealed that down regulation of NPC contributed to a significant reduction of DAG in Arabidopsis under phosphate-deficient conditions (Chapman & Ohlrogge, 2011). The acyl transfer reaction generates DAG which is then catalyzed by DGK enzyme to produce PA. Concurrentlythe expression of OeDGK7 genes were down-regulating in CG. DGKs are lipid kinases that catalyze DAG phosphorylation to generate PA (Sanjaya et al., 2011), which influences the content of DAG. We postulate that the activity of OeDGK and OeDPP1 enzymes influences the content of DAG in olives.

Studies indicated that the TAG biosynthesis, mediated by PDAT1 is involved in transferring FAs from membrane lipids to peroxisomal β-oxidation and plays a role in maintaining membrane lipid homeostasis (Fan et al., 2014). Fan et al. (2013) noted that overexpression of PDAT1 in Arabidopsis leaves resulted in a noticeable accumulation in TAG content. In addition, the experiment revealed that PDAT1 appears to have an intense tendency for PC-containing acyl groups with many double bonds, such as PUFA (Ståhl et al., 2004). Kim et al. (2011) pointed out that the PDAT from castor beans selectively transfers C18:1 to TAG, altering the UFAs content and increasing TAG accumulation. Compared with QJ, the expression of OePDAT genes decreased 5.71-fold in CG, which consistented with the low levels of TAG and C18:1. DGAT is in charge of diverting DAG from membrane lipid biosynthesis into TAG, which activity may have important implications for carbon influx into seed oil in some species (Weselake et al., 2009). Based on the study, overexpression of DGAT in Arabidopsis thaliana seeds effectively increased TAG biosynthesis (Jako et al., 2001). In contrast to the trend of DGAT accumulation in castor seeds (Shockey et al., 2006), its expression was down-regulated 4.72-fold in olive, which had considerably decreased expression than DGAT2 (2.33-fold). Research confirmed that PDAT and DGAT play synergistic roles in catalyzing the TAG biosynthesis in different plants (Zhang et al., 2009). When PDAT1 was inhibited by RNAi-mediated gene silencing under DGAT1-knockout condition, the oil content fell by an additional 63% relative to the DGAT1 control (Zhang et al., 2009). Our experiment showed that the downregulation trend of OePDAT gene was more pronounced than that of OeDGAT and may have a significant impact on DAG accumulation in olives, which has been validated in the study of Arabidopsis thaliana (Fan et al., 2013).Together, our findings suggest that OePDAT1 is more important for TAG biosynthesis in PUFA-rich oil crops than OeDGAT1.

The content of SFA and PUFA in olive oil was primary accumulated in CG, such as C16:0 and C18:2. SAD is the essential enzyme for PUFA biosynthesis in the FAs desaturation pathway, which directly determines the total content of SFA in vegetable oils and the ratio of SFA to UFA (Craig et al., 2008). The study has demonstrated that SAD catalyzed the initial step in the desaturation of SFAs by desaturating stearic acid (C18:0-ACP) into oleic acid (C18:1-ACP) (Pidkowich et al., 2007). Research indicated that the SAD gene was efficiently expressed in tobacco, promoting conversion of C18:0 to C18:1 and enhancing chilling resistance (Craig et al., 2008). The up-regulated expression of SAD in CG promoted the accumulation of C18:1 in plastids compared with QJ. However, we noticed that the content of C18:1 was lower in CG and speculated that C18:1 accumulation was regulated concurrently by several enzymes.

There were two thioesterases in plants, including FATA and FATB. Thioesterases have function in catalyzing hydrolysis of freshly acyl-ACPs to release FFA and ACP into plastid, thereby influencing FAs compositions in some degree (Bonaventure et al., 2003). The selectivity for acyl-ACP was different between these two thioesterases. In contrast to FATA, FATB particularly catalyzes the hydrolysis of saturated acyl-ACPs, such as palmitoyl-ACP (Ohlroggeav & Browse, 1995). Ozseyhan et al. (2018) demonstrated that silencing the FATB genes in camelina with an artificial microRNA targeting gene-specific regions resulted in a 45% reduction in C16:0 and a 38% decrease in C18:0 content, while increasing C18:1 content. In our study, the expression of OeFATB was higher in CG, which is consistent with a higher level of C16:0 and a lower level of 18:1. The omega-6 fatty acid desaturase (FAD2) is an integral membrane protein in the ER, that primarily desaturating C18:1 into C18:2 (Ohlroggeav & Browse, 1995). Liu, Singh & Green (2002) suggested that RNA-mediated FAD2-1 gene silencing resulted in significantly enhancement of C18:1 content, up to 77% in contrast to 15% in untransformed cotton seeds. Unver et al. (2017) noted that the decrease in FAD2 expression and increase in SAD expression were consistent with the accumulation of high levels of C18:1 in olive. Meanwhile, the content of C18:2 in rape seed oil was facilitated by the overexpression of FAD2 genes. In our study, OeFAD2 showed significantly higher expression than OeSAD, promoting the consumption of C18:1 and consistent with a higher content of C18:2 in CG.

The rapid exchange of FAs between PC and the acyl-CoA pool via PC-deacylation and a lysophosphatidylcholine (LPC)-reacylation cycle is a significant component of the acyl editing process. In the forward reaction (LPC-reacylation), LCATs combine acyl groups at both sn-1 and sn-2 positions with a propensity for C18-unsaturated acyl-CoAs. In the reverse reaction, LCATs prefer to hydrolyzes oleoyl-PC at the sn-1 position to produce lysophospholipids (Lager et al., 2013). FAs (PUFA-PC) complete the desaturation process in the plastid and then participate in the Kennedy pathway by releasing linoleoyl-CoA and linolenic-CoA via the reverse action of the LCAT enzyme. In our study, the expression level of OeLCAT was increased in CG, compared with QJ. Therefore, more C18:2 and C18:3 at the sn-2 position of PC catalyzed by OeLCAT could be reserved, thereby maximising the nutritional value of VOO. Olive is an oil crops with specific FAs composition (higher level of C18:1 and little amount of SFA), which is beneficial to human health. C18:2 and C18:3, as essential FAs, are an important nutrientions for daily diet. It is necessary to increase the lower content of PUFA in VOO through molecular breeding technology. However, higher levels of PUFA accelerate the oxidation of olive oil, which is detrimental to storage and sale. Therefore, many factors need to be taken into account to regulate the proportion of FAs.

Conclusions

This study analyzed the factors influencing the DAG accumulation and FAs composition of olive mesocarps by combining transcriptomic and lipidomic analysis. Based on the results and discussion, the following conclusions can be drawn: 1) Transcriptomic analysis identified that, among the five pathways related to glycerolipids biosynthesis, most genes were down-regulated expressed in CG compare with QJ, which were contrary to the trend of DAG accumulation. In particular, OeNPC genes were expressed at higher levels while OeDGK, OeDPP1, OePDAT, OeDGAT1 and OeDGAT2 showed the opposite trend in CG. We speculate that the non-acyl CoA pathway is the main route for DAG accumulation in olive, rather than the traditional Kennedy pathway.

2) C18:2 extensively accumulated in CG, compared with QJ. We found that OeFAD2 gene plays a more important role in promoting C18:2 accumulation than the OeSAD gene. Meanwhile, in the PC-acyl-CoA pool cycle pathway, more C18:2 flow from PC catalyzed by OeLCAT3 to participate in DAG biosynthesis.

For the first time, we analyzed the molecular mechanisms of DAG accumulation and FAs composition in olives selected from China. These results were used to promote genetic improvement or selection to improve the DAG content of VOO and alter the ratio of PUFAs based on hunman health needs. Furthermore, investigating further molecular mechanisms affecting the ratio of DAG isomers (1,2-DAG and 1,3-DAG) is necessary to enhance the nutritional value of olive oil.

Supplemental Information

Supplemental Information 1 Oil content of fresh olive fruit at different maturity stages.

The data are shown as means of three biology repeats ± SD.

Click here for additional data file.

Supplemental Information 2 Composition of lipid class (CG vs QJ).

The different lipid class are represented by different colours and the proportion of them is indicated by the size of the colour block area.

Click here for additional data file.

Supplemental Information 3 Orthogonal partial least squares discriminant analysis (OPLS-DA) of lipidomic.

In the figure t[1] represents principal component one, to[1] represents principal component two, and the ellipse represents the 95% confidence interval. Points of the same colour indicate individual biological replicates within the group, and the distribution status of the points reflects the degree of variation between and within groups.(R2X (cum):0.839, R2Y (cum):0.994).

Click here for additional data file.

Supplemental Information 4 Pearson correlation analysis between CG and QJ.

The horizontal and vertical coordinates in the graph represent the samples and the numbers in the circles represent the correlation coefficients of the two samples. (A) Hierarchical cluster analysis (HCA) analysis was used to determine the variation in differential genes between groups. Color scale represents differential gene expression. Red color represents highly expressed gene and blue color represents low expressed gene (B).

Click here for additional data file.

Supplemental Information 5 The primers used in this study.

Click here for additional data file.

Supplemental Information 6 Abbreviation list of lipids and genes used in this study.

Click here for additional data file.

Supplemental Information 7 Olive oil content.

Click here for additional data file.

Supplemental Information 8 Fatty acid composition and relative content.

Click here for additional data file.

Supplemental Information 9 Lipid Concentration result.

Click here for additional data file.

Supplemental Information 10 Lipid Class result.

Click here for additional data file.

Supplemental Information 11 Quality of sample data and statistics of reads and reference genome comparisons.

Click here for additional data file.

Supplemental Information 12 Differentially expressed genes under different metabolic pathways.

Click here for additional data file.

Supplemental Information 13 Enrichment pathways for differentially expressed genes.

Click here for additional data file.

Additional Information and Declarations

Competing Interests

Author Contributions

Data Availability

The authors declare that they have no competing interests.

Wei Hu conceived and designed the experiments, performed the experiments, analyzed the data, prepared figures and/or tables, authored or reviewed drafts of the article, and approved the final draft.

Junyi Ma conceived and designed the experiments, authored or reviewed drafts of the article, and approved the final draft.

Hongjie Zhang performed the experiments, authored or reviewed drafts of the article, and approved the final draft.

Xin Miu performed the experiments, authored or reviewed drafts of the article, and approved the final draft.

Xin Miao performed the experiments, analyzed the data, authored or reviewed drafts of the article, and approved the final draft.

Yu Deng performed the experiments, authored or reviewed drafts of the article, and approved the final draft.

The following information was supplied regarding data availability:

The data is available in the Supplemental Files and at Mendeley Data:

Ma, Junyi; Hu, Wei (2023), “Integrated lipidomic and transcriptomic analysis reveals diacylglycerol accumulation in olive of Longnan (China)”, Mendeley Data, V1, DOI 10.17632/4k994zmjjk.1.

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
