# Peer review of "Integrated lipidomic and transcriptomic analysis reveals diacylglycerol accumulation in olive of Longnan (China)"

_PeerJ, doi:10.7717/peerj.15724_

## Round 0.1 · original submission · Major Revisions

Your manuscript was reviewed by two experts in the field. The reviewers found the work interesting but raised several issues which need to be addressed properly. The reviewers provide detailed comments in their reviews and point out the areas where the manuscript needs to be improved. I also read the manuscript carefully and largely agree with the reviewers’ comments. In particular, extensive rewriting of this manuscript with a clear hypothesis and language is critically important for publication in PeerJ.

Reviewer 1 ·

Basic reporting

Manuscript no. 83220 is not ready for publication in its current form. The level of revision needed to make it publishable is so high that I do not see any option but to reject it. I will point out major issues with the manuscript so the authors can work on it. I will also indicate positive features that could be salvaged if only better presented.

1. The English used in this manuscript is simply too confusing. Verb tenses and meanings are mixed, which makes for ambiguous sentences and truncated reading.
2. Authors refer to genes and their products interchangeably and do not adhere to gene nomenclature standards.
3. The introduction section does not follow a logical sequence and it fails to expose a clear hypothesis.
4. Acronyms are used indiscriminately. Some are not defined at first mention; some are defined and then never again used; most are outright unnecessary.
5. The figures are aesthetically pleasing, although authors should be mindful of colour schemes and of how presenting all data (twice, in the case of Fig. 4C) is not always the best way to convey information visually.

Experimental design

6. Research questions are vaguely alluded to, but never exactly defined. The field in general is relevant and meaningful.
7. The materials and methods section at times reads like a laboratory protocol. It is great when authors include details that can increase reproducibility, but when the text is as confusing as this, that is not accomplished.
8. Authors use univariate and multivariate analyses. At no point is the reasoning behind the choice of analysis presented or a general statistical pipeline mentioned. The number of biological replicates is only mentioned for lipidomics, not for transcriptomics.

Validity of the findings

9. The results section reads like a Biochemistry textbook. It is overly verbose and fails to highlight the main findings.
10. Findings are often misrepresented. The authors studies lipids and transcripts on a global scale. However, that does not automatically translate into factual information about the proteins encoded by those genes and involved in the synthesis of those lipids. The conclusion section, for instance, mentions "increased enzyme activity". The methods section does not include enzyme activity assays. I believe this stems from a fundamental misunderstanding of the levels of biological complexity and of the limitations of the techniques used.

·

Basic reporting

The MS reports the Integrated lipidomic and transcriptomic analysis of diacylglycerol accumulation in two different cultivars of olive mesocarp.
Though results presented are well consistent with those previously published with other crops, I would like to comment (to give recommendations) on the work as follows:
1. Authors should be more to emphasize manuscript novelty.
2. The manuscript contains several writing errors that need to be corrected. Here I am mark up some of these errors, as most can be found with adequate proofreading.

Experimental design

64:“originates”
65: City …Remove “is”
66: Base “is”
67: which accounts for…28% of “the”
69-70: rephrase the sentence and provide a reference.
70-72: cite a reference.
76: higher than “the”
77: “values of approximately “5.5%....
77-79: rephrase the sentence.
86: molecular “mechanisms”
88: references for soybean and rapeseed are not cited
89: “and” physiological regulation
104: remove “Our”
104-107: rephrase sentences.

125: “Lipids were extracted using the following methods.”
128: Change the word “Sitting still”
151: Java Treeview software “was”
152: ….clustering analysis, as well as…
168: abbreviations are missing, abbreviate when use fist in the text for example, GO and line 190: “VOO”
191-193: rephrase the sentence.
202: stability of “the”
211-212: rephrase the sentence.
213: what are those main unsaturated fatty acids? this heading is incomplete rephrase it
215: consequences? I am wondering did you treat the samples? if yes write those conditions.
216: striking replace with “significant.”
219:replace “ accumulation” with “accumulated”
251: remove “etc” and write what other compounds you studied
253: replace “concentrated” with “studied” and who identified these 161 DEGs associated with oil accumulation?
256-261: change to discussion section or rephrase sentences.
265: cite the figure
268: remove “ comma (,) after Pyruvate and pathway
273: what are those four upregulated genes? Missing the link between QJ and CG rephrase this sentence.
274: Since authors have studied gene expression only, I am wondering if they also measured the pyruvate. If so, how much of it is being converted into CO2 and H2O?
278-283: remove sentences and cite these references in discussion section but not in results.
322: replace with “results were compared”
327: replace “ discovered” with observed or found
340: Italic “Brassica napus”
349: where is the results for increased oil content in QJ?
371: change down-regulated expression to “down regulation of “
392: Italics “Arabidopsis thaliana”
401: Italics “Arabidopsis thaliana” check all gene and plant scientific names in italics
407: cite a reference
409-411: rephrase the sentence.
414-416: cite a reference
453: write the factors you studied

Validity of the findings

no comment

Additional comments

no comment

---

## Round 0.2 · accepted · Accept

I appreciate the authors' efforts in revising the manuscript substantially. The current version is suitable for publication in PeerJ.

·

Basic reporting

The authors have have satisfactorily addressed all my comments, I recommend this article to be accepted for publication

Experimental design

N/A

Validity of the findings

N/A

Additional comments

N/A